# Microwave Treatment for Citrus Huanglongbing Control: Pathogen Elimination and Metabolomic Analysis

**DOI:** 10.3390/plants14172712

**Published:** 2025-09-01

**Authors:** Xianrui Chen, Yunyun Li, Gen Li, Yanling Wu, Junru Mao, Jiasheng Lin, Mengxue Diao, Zhimin Huang

**Affiliations:** National Key Laboratory of Non-Food Biomass Energy Technology, Guangxi Key Laboratory of Advanced Microwave Manufacturing Technology, Guangxi Academy of Sciences, 98 Daling Road, Nanning 530007, China; lyy00795@hnu.edu.cn (Y.L.); ligenbio@126.com (G.L.); wuyanling1108@163.com (Y.W.); maojunru0505@163.com (J.M.); linjs07@163.com (J.L.); mengxuediao@gxas.cn (M.D.)

**Keywords:** Huanglongbing, *Candidatus* Liberibacter asiaticus, microwave treatment, plant stress response, metabolomics

## Abstract

Huanglongbing (HLB), associated with *Candidatus* Liberibacter asiaticus (*C*Las), has severely impacted global citrus production, with no economically viable control measures currently available. This study explored microwave treatment at 2450 MHz as an innovative physical method for HLB control, combining pathogen elimination efficacy with metabolomic analysis. In controlled experiments, 36 HLB-infected citrus plants were treated with 500 W or 250 W microwave irradiation and underwent 10 cycles, achieving up to 99.83% reduction *C*Las titer. Non-targeted metabolomic analysis identified 15 significantly altered metabolites, including upregulated beta-caryophyllene and lysophosphatidylinositols, and downregulated 5′-S-methyl-5′-thioadenosine. The results indicate that microwave treatment effectively suppressed *C*Las while simultaneously triggering citrus physiological metabolic changes. These findings suggest that microwave treatment could serve as a sustainable alternative to chemical controls. However, further optimization of parameters, such as wavelengths, voltages, currents, and safety protocols, will be essential for practical field implementation.

## 1. Introduction

Citrus is one of the most widely cultivated fruit crops globally, grown in over 130 countries across tropical and subtropical regions [1,2]. However, its yield and quality have long been threatened by various pathogenic microorganisms, among which Huanglongbing (HLB) is recognized as the most devastating citrus disease [3,4]. HLB is associated with phloem-limited bacteria of the genus *Candidatus* Liberibacter [5,6], which includes three confirmed species: *Candidatus* Liberibacter asiaticus (*C*Las), *Candidatus* Liberibacter africanus (*C*Laf), and *Candidatus* Liberibacter americanus (*C*Lam) [7,8]. Over the past decade, HLB has spread across major citrus-producing regions worldwide, causing up to 100% yield loss in parts of Africa [9,10], a 38% infection rate in São Paulo State, Brazil [11], and a 95% yield reduction in Florida, according to the final USDA estimate for the 2024–2025 season orange crop. In China, the southern citrus-growing provinces have also been severely impacted [12]. The challenges in controlling HLB are compounded by the inability to culture its pathogen in vitro, along with its long incubation period, the presence of asymptomatic early stages, and difficulties in diagnosis. These factors significantly hinder early detection and effective management of the disease.

Multiple complementary strategies, including resistance breeding, biological, chemical, and physical interventions have been widely applied for HLB control. Regarding genetic resistance, transgenic citrus plants overexpressing antimicrobial peptides like thionin, cecropin B, and attacin A have demonstrated significant resistance to HLB [13,14,15]. Additionally, techniques such as somatic hybridization, polyploid breeding, and natural field selection have led to the development of several HLB-tolerant plant varieties [16]. Biological control strategies utilizing Bacillus-based agents have been shown to improve soil microecology and reduce the risk of pathogen transmission [17]. Moreover, AI-assisted drug screening has also demonstrated potential to clear the pathogen and enhance fruit quality [18]. *C*Las shows thermal sensitivity and reduced viability at temperatures above 35 °C [19]. Physical control strategies, particularly heat-based techniques, including moist hot steam, infrared heating, and hot water treatment, have been employed to target and eliminate the *C*Las [20]. These thermal therapies capitalize on the temperature differential between pathogen and host to effectively suppress HLB spread without significantly harming the citrus plants. However, up to this point, none of the attempts have proven to be economically viable.

Due to the limited effectiveness of current HLB management strategies, this study tested microwave treatment to see if it could be a promising physical approach. Operating within the 300 MHz–300 GHz spectrum, microwave offers advantages such as rapid heating, low energy consumption, and residue-free application [21]. Antimicrobial activity results from both thermal effects and non-thermal mechanisms, with the latter primarily due to disruption of cellular structures [22]. The thermal effect of the microwave arises from the interaction between the alternating electric field and polar molecules within the citrus tissue, generating heat through molecular friction [23]. Due to its penetrating nature, microwave heating ensures uniform temperature distribution both internally and externally. Additionally, microwave exhibits non-thermal effects, where electromagnetic energy alters the charge distribution and transmembrane potential of cell membranes, leading to membrane rupture and lysis [24]. Furthermore, the high-frequency oscillation of microwave can also disrupt macromolecules such as proteins and nucleic acids, which may induce cell death. [25]. Recent studies explored the application of microwave in the treatment of agricultural soil [26]. In the realm of plant disease management, microwave irradiation was shown to effectively eliminate endophytic fungi in *Achnatherum inebrians* and *Elymus dahuricus* [27,28]. Preliminary findings further indicate that a 90-day microwave treatment reduced the levels of *C*Las in *Catharanthus roseus* by 99.98% [29], suggesting strong potential for HLB suppression.

This study systematically investigated the bactericidal effect of microwave treatment on the HLB-associated *C*Las. Key parameters, including frequency, power, exposure duration, and temperature, were optimized to evaluate the elimination efficacy of *C*Las during the study. Metabolomic and chemometric analyses were employed to assess the impact of the microwave effects on the citrus metabolism. The pathway of significant differential metabolites in citrus was analyzed to clarify the physiological responses under combined thermal and electromagnetic stress conditions. This work provides a theoretical foundation and technical support to assess the overall feasibility and application potential of microwave-based HLB control.

## 2. Results

### 2.1. Detection of Candidatus *Liberibacter* Asiaticus in Citrus

Citrus seedlings used in the experiment were transplanted into planting pots and cultivated in a sunlit nursery. After 30 days of cultivation, DNA was extracted from the citrus leaves and screened for *C*Las by the quantitative real-time PCR (qPCR) method (Figure 1a). A standard curve was created by plotting the logarithm of plasmid pUC-CLas16S copy numbers against the corresponding qPCR *C*_T_ values: y = 3.5063x + 7.4913, where “y” represents the *C*_T_ value, “x” corresponds to the log(DNA concentration), *R*^2^ = 0.9992, allowing the absolute quantification of *C*Las in the experimental samples. The results indicated that qPCR detection confirmed the presence of *C*Las in a number of citrus plants, some of which exhibited noticeable yellowing of leaves (Figure 1b), one of the typical symptoms of HLB. Subsequently, the *C*Las-infected citrus plants were cultivated separately under psyllid-proof nets to prevent cross-infection.

### 2.2. The Tolerance of Citrus to Microwave

A fully digital industrial microwave system (Figure 2a) operating at a frequency of 2450 MHz and a power range from 250 W to 1500 W, was utilized for the microwave treatment of citrus plants (Figure 2b). Each citrus plant received a single treatment lasting between 15 and 30 s. The temperature was monitored in real-time using an infrared thermal imaging device (Figure 2c), recording the highest temperatures reached during treatment. Following the microwave treatment, the citrus plants were cultured for 90 days and categorized into three grades based on their growth activity. Grade 1: Undamaged—the microwave treatment was within the tolerance range of citrus, and the plants remained healthy with no visible physiological harm (Figure 3a). Grade 2: Damaged—due to the thermal and non-thermal effects of the microwave; these citrus plants showed partial wilting of the leaves and stems (Figure 3b). Grade 3: Dead—these citrus plants exhibited severe dehydration and desiccation due to excessive microwave-induced stress, leading to irreversible damage and death (Figure 3c).

The experimental results (Table 1) showed that citrus plants treated with microwave at power levels between 1000 W and 1500 W suffered damage or death. This indicates that the microwave energy was too intense for the citrus plants, causing a rapid temperature rise and severe dehydration within a short period. The highest temperature recorded exceeded 56 °C, which resulted in the death of the plants. Citrus plants subjected to microwave treatment at 500 W for 25–30 s experienced damage, while a shorter exposure of 15 to 20 s, resulting in a peak temperature below 54 °C, helped preserve the health of the plants with no damage observed. Lowering the microwave power to 250 W and extending the duration to 15 to 40 s did not harm the seedlings. To ensure the healthy growth of citrus fruits, the maximum temperature for microwave processing should not exceed 52 °C. As a result, the selected microwave treatment conditions for eliminating *C*Las were found to be 500 W for 15 to 20 s and 250 W for 15 to 30 s.

### 2.3. Microwave Treatment for the Elimination of CLas

In comparison to the control group (which did not receive microwave treatment, citrus numbers N1–N6), the *C*Las titers in citrus subjected to microwave treatment (citrus numbers M1–M36) showed different degrees of reduction after 90 days of cultivation. The percentage change of *C*Las titer ranged from 18.10% to 99.83% (Table 2). Specifically, the average percentage changes of *C*Las titer in citrus treated with 500 W for 20 s and 15 s were 89.51% and 79.86%, respectively. For those treated at 250 W, the average percentage changes of *C*Las titer for 30 s, 25 s, 20 s, and 15 s were 70.40%, 57.01%, 39.50%, and 26.25%, respectively. These results demonstrated that microwave treatment exhibited significant effects on the elimination of *C*Las in citrus phloem.

Among the microwave-treated plants, citrus M2 showed the highest efficiency in eliminating *C*Las. The cycle threshold (*C*_T_) value of qPCR increased from 19.4 to 29.1, corresponding to a reduction in the titer of *C*Las from 2.01 × 10^7^ to 3.44 × 10^4^ copies per ng of DNA extracted from citrus leaves. This represented a remarkable decrease of 99.83% in the percentage change of *C*Las titer. In contrast, citrus M34 exhibited relatively low efficiency in eliminating *C*Las, with the *C*_T_ value changing slightly from 31.8 to 32.7. The pathogen load for M34 decreased from 3.28 × 10^3^ to 2.69 × 10^3^ copies per ng of DNA, resulting in an 18.10% reduction.

In the control group, the titers of *C*Las increased significantly after 90 days of cultivation, showing an average 682.63% rise in pathogen concentration. This observation confirmed that *C*Las proliferated rapidly in citrus plants without any intervention. The most notable increase in pathogen levels was found in citrus N1, where the *C*_T_ value of *C*Las decreased from 32.2 to 27.4. The *C*Las titer rose from 4.49 × 10^3^ to 1.05 × 10^5^ copies/ng of DNA, reflecting a remarkable increase of 2238.65% compared to day 0. In contrast, citrus N6 exhibited only mild proliferation of *C*Las, with the *C*_T_ value changing slightly from 23.8 to 23.1, leading to an increase in *C*Las titer from 1.12 × 10^6^ to 1.77 × 10^6^ copies/ng of DNA, which represents a 58.36% rise.

### 2.4. The Effect of Microwave Treatment on the Metabolites of Citrus

The citrus samples M1–M6 from the microwave treatment group (Table 2, 500 W, 20 s), which demonstrated the highest efficacy in eliminating *C*Las, along with the control group citrus samples N1–N6, were utilized for non-targeted metabolomic analysis. A total of 1747 metabolites were identified in both the positive and negative polarity modes of mass spectrometry (Table 3). Groups A, C, E, G, and J represent the citrus plants M1 to M6, which were cultivated for 0 h, 24 h, 48 h, 72 h, and 96 h after being subjected to microwave treatment. Groups B, D, F, H, and K represent the citrus plants N1 to N6, which were cultivated for the same durations without any treatment.

Differential metabolites were screened based on a significance threshold of *p* < 0.05. In the A vs. B comparison group, a total of 160 metabolites exhibited significant differences across both polarity modes, with 74 metabolites upregulated and 86 downregulated. In the C vs. D group, 123 differential metabolites were identified, comprising 31 upregulated and 92 downregulated. The E vs. Fd comparison revealed the largest number of significantly altered metabolites, totaling 174, with 43 upregulated and 131 downregulated. For the G vs. H group, 95 metabolites showed significant variation, including 40 upregulated and 55 downregulated. Lastly, in the J vs. K comparison, the smallest number of differentially expressed metabolites was observed, with 59 total, encompassing 28 upregulated and 31 downregulated.

As shown in Figure 4, a total of 296 differential metabolites were identified across all comparison groups using positive polarity mode, with only two metabolites commonly altered in all groups. Among the comparisons, the G vs. H group displayed the least metabolic divergence, featuring 17 unique differential metabolites. In contrast, the E vs. Fd group exhibited the greatest divergence, with 71 unique differential metabolites. In negative polarity mode, 164 differential metabolites were identified, of which three were common across all groups. The J vs. K group showed the smallest number of unique differential metabolites (*n =* 2), indicating minimal metabolic variation. Conversely, the A vs. B group had the largest number of unique differential metabolites (*n* = 37), reflecting a more substantial metabolic shift.

A total Number of 15 differential metabolites were found to be shared among four or more comparison groups, as summarized in Table 4. Among these metabolites, 12 were upregulated, while three were downregulated. Specifically, six of the metabolites were associated with the category of lipids and lipid-like molecules, with three being upregulated and three downregulated. Additionally, two metabolites belonging to the organic acids and derivatives class were both downregulated. One metabolite, which falls under the category of nucleosides, nucleotides, and analogues, was also downregulated. The remaining six downregulated metabolites are currently unclassified in the reference database, indicating that they may be potential novel or yet-to-be-characterized metabolic compounds. These shared differential metabolites may represent core metabolic responses to microwave treatment and warrant further investigation for their potential biological significance.

## 3. Discussion

The citrus HLB pathogen *C*Las resides in the phloem of citrus plants and is primarily transmitted by the insect vector citrus psyllid [30,31]. To control citrus psyllids, citrus farms commonly rely on chemical pesticides such as chlorpyrifos and imidacloprid. This approach has been widely adopted in HLB-endemic regions of southern China. However, the extensive use of chemical agents has frequently resulted in increased resistance among psyllids, creating long-term challenges for disease management [32]. Studies have shown that applying electric current can be effective in controlling Weligama Coconut Leaf Wilt Disease [33], and using a weak electromagnetic field to manage the health of lime plants infected by Candidatus *Phytoplasma aurantifoliae* has also yielded positive results [34]. Microwave treatment, as a physical sterilization method, offers several advantages, including absence of chemical residues and environmental safety. These properties make it a promising alternative for the elimination of HLB *C*Las. Table 2 presents the percentage change in *C*Las titer following treatment. This variation might be attributable to the differences in the plants’ sensitivity to HLB, as well as the differing concentrations of pathogens present before treatment. Higher temperatures and longer durations during microwave treatment yield better pathogen elimination.

When using steam heat treatment to target *C*Las in citrus, the highest surface temperature recorded for the plants was 60 °C, which did not cause significant harm [35]. In this study, microwave treatment was applied to citrus. It was found that temperatures of 52 °C or lower did not damage the plants, while temperatures above 53 °C resulted in significant damage and those exceeding 56 °C were lethal. Therefore, the key factor in using microwaves to eliminate *C*Las is to ensure a balance between the lethal temperature for the pathogen and the tolerance temperature of the citrus.

Earlier studies suggested a greenhouse-based heat treatment method for citrus HLB, where citrus plants were heated to 48 °C in a controlled environment, achieving an average 82.28% reduction in *C*Las titer after 30 days of treatment [36]. In this study, microwave treatment at 500 W raised the highest temperature of the citrus plants to 52 °C, which is the lethal threshold for *C*Las in the phloem. However, a single microwave application proved insufficient to eliminate most of the *C*Las within the plant due to the short exposure time and limited energy accumulation. To improve the effectiveness, a repeated microwave treatment protocol was implemented: after allowing the citrus plants to return to ambient temperature, the microwave exposure was repeated a total of 10 cycles. This approach achieved the highest 99.83% reduction rate of *C*Las titer in HLB citrus. In comparison, conventional heat treatments, such as warm water immersion at 45 °C (requiring 12–24 h) or incubation in a growth chamber at 40–42 °C (taking over 48 h), demanded significantly longer durations to eliminate *C*Las [37,38]. These findings underscore the superior efficiency of microwave treatment, which benefits from the synergistic combination of thermal and non-thermal effects. Supporting evidence came from an experiment where HLB-infected periwinkle plants (*Catharanthus roseus*) were subjected to microwave-induced heat treatment (48 °C, 26–94 s), and the *C*Las titer in the plants decreased by 99.98% after 90 days [29]. This effective microwave heating method ensures an even temperature increase across both external and internal tissues, including the phloem, greatly enhancing the efficiency of *C*Las elimination.

To achieve a 100% microwave elimination efficiency for *C*Las and to ensure that no hidden bacteria re-emerge after treatment, further work is needed, including experiments using various microwave frequencies, voltages, currents, and citrus cultivars. Additionally, the microwave antenna’s radiation pattern is a variable that can be investigated. Ideally, the equipment should be able to treat both the aboveground and underground parts of the plants simultaneously. This will require the development of new microwave devices equipped with highly intelligent control systems to ensure the adjustability of the microwave parameters and coverage. The authors believe that with the introduction of more effective microwave technology, researchers will be able to conduct further experiments and provide more data on efficacy. This includes the reduction in *C*Las titer and the prolonged health of the citrus tree.

Citrus plants activate their defense and repair systems in response to external stress, which leads to changes in their physiological and biochemical activities, resulting in variations in metabolite content. [39,40,41]. In this study, microwave treatment for the elimination of *C*Las imposed both thermal and non-thermal stresses on citrus. Metabolomic analysis revealed that 15 were differentially expressed in the microwave-treated citrus groups, showing either upregulation or downregulation. These changes in metabolite levels indicate that specific pathways may be involved in the citrus response and repair mechanisms following microwave treatment.

This study analyzed several differentially expressed metabolites related to cellular damage and repair mechanisms. Beta-caryophyllene (Table 4, Num. 1) is a compound whose biosynthesis is catalyzed by sesquiterpene synthases, such as caryophyllene synthase. In microwave-treated citrus, levels of beta-caryophyllene were found to be significantly elevated. The expression of these synthases is influenced by environmental stress and hormonal signals, which may help mediate defense responses by modulating the pathways of jasmonic acid and salicylic acid [42]. Two species of lysophosphatidylinositol (LPI) species—LPI 16:0 and LPI 18:3 (Table 4, Num. 2 and Num. 3)—showed significant accumulation following microwave treatment. These LPI metabolites are likely produced through the hydrolysis of phosphatidylinositol by phospholipase, such as phospholipase A2. They have been linked to stress responses and developmental regulation [43]. LPIs may play a role in stress adaptation by modulating protein kinase (e.g., MAPK) or calcium signaling pathways, influencing ion channel activity, maintaining membrane integrity, and facilitating lipid remodeling [44,45]. Notably, 5′-S-methyl-5′-thioadenosine (MTA) displayed downregulated expression in treated citrus (Table 4, Num. 9). MTA is a metabolic product of S-adenosylmethionine and is involved in the methionine salvage pathway. This process occurs through the action of either MTA phosphorylase or MTA nucleosidase, which ultimately regenerates methionine and adenosine. This regeneration affects the availability of nucleotide precursors and, in turn, influences the synthesis of DNA and RNA [46].

The findings indicate that microwave treatment triggers complex metabolic responses in citrus plants. These responses include the activation of pathways related to defense-related secondary metabolites, the remodeling and repair of membrane lipids, and a temporary suppression of nucleotide metabolism. By analyzing the trends in the number of differential metabolites across various comparison groups of citrus (as shown in Table 3), this study suggests that the impact of microwave treatment on citrus metabolism peaks at 48 h after treatment and gradually diminishes by 96 h.

Citrus fruits are sweet, and people love them. The authors hope that farmers will feel extremely proud of their good harvest and the quality of the citrus fruits. Agricultural researchers from China and other countries are working diligently to help citrus crops thrive once again. Whether researchers use microwave treatment or other methods to control HLB, a long-term strategy for eliminating *C*Las also needs to be combined with effective control measures against citrus psyllids; the ultimate goal is to protect and support the citrus industry.

## 4. Materials and Methods

### 4.1. Plant Materials

Two-year-old citrus plants of the same variety, navel orange (*Citrus sinensis* Osb. var. *brasliliensis* Tanaka), cultivated in Hezhou City China, were selected for this study. These plants, which had a height range of 50 to 60 cm, exhibited optimal growth conditions and showed no signs of mechanical damage. The citrus plants were subsequently used to investigate the effects of microwave radiation on the elimination of *C*Las and on the metabolomics.

### 4.2. Microwave Treatment of Citrus Plants

The microwave treatment of citrus plants was performed using a WeboX-A6 fully digital industrial microwave system (Weilang Technology Co., Ltd., Zhuzhou City, China) operating at a frequency of 2450 MHz with an adjustable power output ranging from 150 W to 5100 W. During the microwave treatment experiment, citrus plants were placed in the microwave resonance chamber. The microwave irradiation was then initiated, and real-time temperature monitoring was conducted using a thermal imaging camera to record the surface temperature distribution of the citrus throughout the treatment.

To determine the microwave tolerance of citrus plants, treatments were given in 16 groups: (1) Power levels of 250 W, durations of 15 s; (2) 250 W, 20 s; (3) 250 W, 25 s; (4) 250 W, 30 s; (5) 500 W, 15 s; (6) 500 W, 20 s; (7) 500 W, 25 s; (8) 500 W, 30 s; (9) 1000 W, 15 s; (10) 1000 W, 20 s; (11) 1000 W, 25 s; (12) 1000 W, 30 s; (13) 1500 W, 15 s; (14) 1500 W, 20 s; (15) 1500 W, 25 s; (16) 1500 W, 30 s. Each group of citrus plants that underwent microwave treatment consisted of six trees.

Microwave treatments to reduce or eliminate *C*Las in citrus were given in six groups: (1) Power levels of 250 W, durations of 15 s; (2) 250 W, 20 s; (3) 250 W, 25 s; (4) 250 W, 30 s; (5) 500 W, 15 s; (6) 500 W, 20 s. In each of the microwave experiments, one cycle of microwave treatment was completed, then the machine was paused to allow the plants to return to room temperature (23 ± 2 °C). Once the trees had cooled, the machine was restarted and the process for another cycle was repeated. The total treatment required ten cycles, with cooling pauses between each successive treatment.

### 4.3. Real-Time Fluorescence Quantitative PCR (qPCR) Detection of CLas

Leaf samples were collected from the upper, middle, and lower canopy positions of each citrus plant. The mesophyll tissue was carefully removed, leaving only the main veins, which were then cut into 0.2–0.5 cm segments. These vein segments were immediately placed in liquid nitrogen and ground into fine powder using a mortar and pestle. The homogenized powder was transferred to 2 mL microcentrifuge tubes for subsequent DNA extraction. Total DNA was extracted using a broad-spectrum plant DNA extraction kit (Meiji Biotechnology Co., Ltd., Guangzhou city, China) following the manufacturer’s protocol. The extracted DNA was quantified and assessed for purity using spectrophotometry prior to quantitative PCR (qPCR) analysis.

The q-PCR detection was performed using 7500 fast Real-Time PCR system (Applied Biosystems, Carlsbad, CA, USA) and Premix Ex Taq (Probe qPCR) kit (TaKaRa). The primers used were as follows: forward primer HLBas: 5′-TCGAGCGCGTATGCAATACG-3′, reverse primer HLBr: 5′-CGTTATCCGTGTAGAGAGGTG-3′, and Probe primer HLBProbe: 5′-(FAM)-AGACGGGTGAGTAACGCG-(MGB)-3′ [47]. The primers were commercially synthesized by GenScript Biotech Corporation (Wuhan, China) with HPLC purification. The q-PCR reaction procedure was as follows: pre-denaturation at 95 °C for 30 s; {PCR denaturation at 95 °C for 5 s, reaction at 60 °C for 34 s} × 40 cycles.

A standard curve was created based on the positive control template of DNA concentration and the *C*_T_ value of q-PCR detection. The 1171-bp fragment of *C*Las 16S rDNA was cloned into the pUC57 plasmid, resulting in a recombinant plasmid of 3887-bp (designated pUC-*C*Las16S) as the positive control template. The pUC-*C*Las16S was quantified and diluted to a working concentration of 100 ng/μL, followed by serial ten-fold dilutions (10^−1^ to 10^−8^ ng/μL) to establish an 8-point calibration series. Each dilution was subjected to qPCR amplification.

### 4.4. Metabolomics Analysis

Leaf samples were collected at 0, 24, 48, 72, and 96 h following microwave treatment for metabolomic analysis. The extraction protocol was performed as follows: (1) A total of 100 mg of liquid nitrogen-ground tissue was transferred to a 1.5 mL microcentrifuge tube, and 500 μL of 80% methanol aqueous solution (*v*/*v*) was added. (2) The mixture was vortexed vigorously for 30 s, incubated on ice for 5 min, and then centrifuged at 15,000× *g* for 20 min at 4 °C. (3) A 200 μL aliquot of the supernatant was taken and diluted with MS-grade water to achieve a final methanol concentration of 53%. (4) The diluted extract was recentrifuged under identical conditions (15,000× *g*, 20 min, 4 °C), and the resulting supernatant was transferred to LC–MS vials for analysis.

The LC–MS analysis was conducted using the following mass spectrometry parameters: mass range—*m*/*z* 100–1500 in full scan mode; ion source (ESI) settings, spray voltage—3.5 kV (positive/negative switching), sheath gas flow rate—35 psi, auxiliary gas flow rate—10 L/min, capillary temperature—320 °C, S-lens RF level—60%, auxiliary gas heater temperature—350 °C; fragmentation—data-dependent MS/MS acquisition (top N = 10) with isolation window 1.6 *m*/*z*, normalized collision energy—30 eV, and dynamic exclusion—15 s.

For metabolomic data processing and metabolite identification, raw mass spectrometry data files were processed using CD3.3 software for metabolite profiling. Molecular formulas were predicted based on molecular ion peaks and fragment ion patterns, followed by database matching against mzCloud (https://www.mzcloud.org/), mzVault, and Masslist to annotate putative metabolites. For quantification, relative peak areas were calculated, and any compounds with a coefficient of variation (CV) >30% in quality control (QC) samples were excluded to ensure data reliability. The final dataset comprised identified metabolites along with their relative quantitative values. All data processing and statistical analyses were performed on a Linux operating system (CentOS 6.6) using R and Python scripts for computational workflows.

The metabolite detection using LC–MS, along with the statistical processing of metabolomic data and the production of publication-quality figures, was supported by Novogene Co., Ltd. (Beijing, China).

## 5. Conclusions

Controlling citrus HLB remains a global challenge. This study demonstrated that microwave treatment can effectively eliminate *C*Las in citrus plants, presenting a promising approach for HLB management. Metabolomic analysis revealed notable changes in specific metabolites, indicating that microwave treatment triggers stress responses while also activating the plant’s defense mechanisms. Future research will take an integrated approach that combines various microwave wavelengths, voltages, currents, and different cultivars of citrus, and authors will also examine the effects of microwave treatment on *C*Las in roots in the field. Equally important is the need to test the electromagnetic shielding effect of the microwave system to ensure user safety. Following this, research should be conducted on achieving good yields and high-quality fruit from citrus trees to evaluate whether microwave treatment could be a viable method for managing HLB. This would help ensure that citrus orchards can continue to provide healthy and delicious citrus fruit products.

## Figures and Tables

**Figure 1 plants-14-02712-f001:**
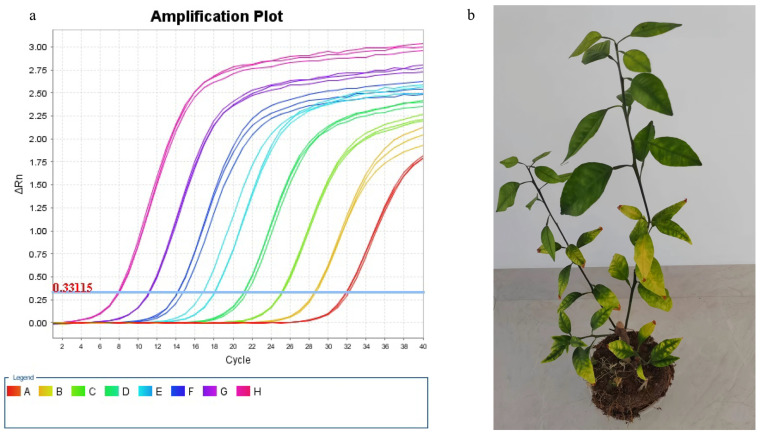
Detection of *C*Las in citrus plants. (**a**) Amplification plots of plasmid pUC-*C*Las16S serial dilutions (10^−1^ to 10^−8^ ng/μL) in qPCR, A: 10^−8^ ng/μL of DNA template, B: 10^−7^ ng/μL of DNA template, C: 10^−6^ ng/μL of DNA template, D: 10^−5^ ng/μL of DNA template, E: 10^−4^ ng/μL of DNA template, F: 10^−3^ ng/μL of DNA template, G: 10^−2^ ng/μL of DNA template, H: 10^−1^ ng/μL of DNA template; (**b**) appearance of citrus plants infected with *C*Las, confirmed by qPCR.

**Figure 2 plants-14-02712-f002:**
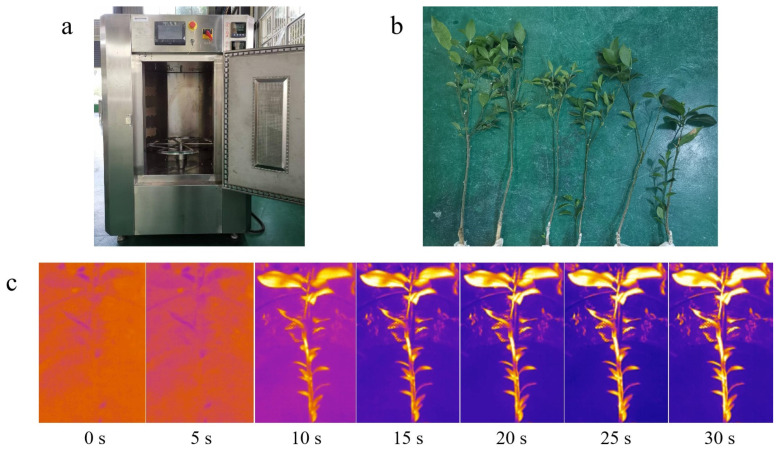
Microwave treatment experiment for citrus. (**a**) A fully digital variable frequency industrial microwave system; (**b**) citrus samples used for the microwave treatment experiment; (**c**) real-time infrared imaging of citrus during a 500 W microwave treatment lasting 30 s.

**Figure 3 plants-14-02712-f003:**
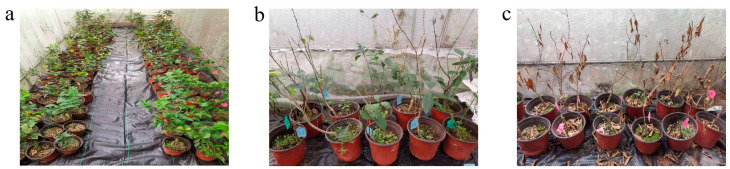
Citrus grading following a 90-day microwave tolerance test. (**a**) Grade 1: Undamaged; (**b**) Grade 2: Damaged; (**c**) Grade 3: Dead.

**Figure 4 plants-14-02712-f004:**
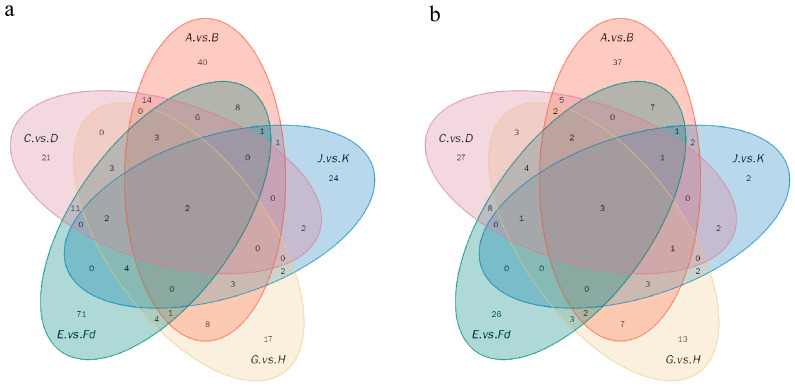
Venn diagram of differential metabolites. (**a**) Differential metabolites across all comparison groups in positive polarity mode [*m*/*z* (+)]; (**b**) Differential metabolites across all comparison groups in negative polarity mode [*m*/*z* (−)].

**Table 1 plants-14-02712-t001:** The tolerance of citrus trees to varying levels of microwave energy.

Citrus Group	Microwave Power (W)	Duration (s)	Highest Temperature (°C)	Citrus Grade * After 90 Days
1	1500	30	62	3
2		25	59	3
3		20	57	3
4		15	54	2
5	1000	30	58	3
6		25	56	3
7		20	54	2
8		15	53	2
9	500	30	55	2
10		25	54	2
11		20	52	1
12		15	50	1
13	250	30	45	1
14		25	40	1
15		20	36	1
16		15	32	1

* Grade 1: Undamaged; Grade 2: Damaged; Grade 3: Dead.

**Table 2 plants-14-02712-t002:** The comparison of *C*Las titer between microwave-treated citrus and a control group of citrus.

Citrus Num.	Treatment		Before Treatment		After Treatment		Percent Change of *C*Las Titer (%) **
			*C*_T_ of *C*Las	Titer of *C*Las	*C*_T_ of *C*Las	Titer of *C*Las	
M1	Microwave treatment *	500 W, 20 s	25.9	2.81 × 10^5^	33.8	1.57 × 10^3^	−99.44
M2			19.4	2.01 × 10^7^	29.1	3.44 × 10^4^	−99.83
M3			30.5	1.37 × 10^4^	34.3	1.13 × 10^3^	−91.75
M4			27.6	9.20 × 10^4^	34.8	8.14 × 10^2^	−99.12
M5			32.3	4.20 × 10^3^	33.9	1.47 × 10^3^	−65.03
M6			31.5	7.11 × 10^3^	34.1	1.29 × 10^3^	−81.87
M7		500 W, 15 s	21.2	2.73 × 10^4^	28.5	1.83 × 10^3^	−93.29
M8			16.8	4.74 × 10^7^	25.4	1.50 × 10^6^	−96.84
M9			31.6	8.52 × 10^3^	33.6	3.34 × 10^3^	−60.80
M10			25.1	1.71 × 10^5^	31.7	2.02 × 10^4^	−88.21
M11			32.1	2.84 × 10^3^	33.5	1.26 × 10^3^	−55.47
M12			28.3	5.62 × 10^4^	32.2	8.69 × 10^3^	−84.53
M13		250 W, 30 s	32.2	2.33 × 10^3^	33.1	1.34 × 10^3^	−42.38
M14			17.9	5.45 × 10^6^	24.4	8.55 × 10^5^	−84.31
M15			28.0	6.17 × 10^4^	32.6	1.39 × 10^4^	−77.52
M16			30.7	4.39 × 10^3^	32.2	1.63 × 10^3^	−62.94
M17			20.6	7.69 × 10^5^	26.3	1.40 × 10^5^	−81.78
M18			25.9	5.38 × 10^4^	31.8	1.43 × 10^4^	−73.47
M19		250 W, 25 s	18.4	2.24 × 10^6^	23.3	5.92 × 10^5^	−73.57
M20			31.2	5.27 × 10^3^	32.4	3.15 × 10^3^	−40.27
M21			28.3	3.10 × 10^4^	31.0	1.17 × 10^4^	−62.39
M22			24.2	2.77 × 10^5^	28.8	8.67 × 10^4^	−68.71
M23			25.5	1.48 × 10^5^	29.3	5.84 × 10^4^	−60.54
M24			32.3	6.22 × 10^3^	33.1	3.95 × 10^3^	−36.55
M25		250 W, 20 s	32.1	6.54 × 10^3^	32.8	4.68 × 10^3^	−28.41
M26			28.1	4.54 × 10^4^	30.0	2.56 × 10^4^	−43.54
M27			26.1	2.93 × 10^5^	28.5	1.52 × 10^5^	−48.26
M28			25.4	2.16 × 10^5^	27.1	9.97 × 10^4^	−53.86
M29			32.5	4.47 × 10^3^	33.1	3.42 × 10^3^	−23.43
M30			25.6	2.94 × 10^5^	28.2	1.81 × 10^5^	−39.50
M31		250 W, 15 s	27.1	1.08 × 10^5^	28.4	8.07 × 10^4^	−25.28
M32			30.9	3.31 × 10^3^	31.4	2.62 × 10^3^	−20.94
M33			24.5	7.64 × 10^5^	26.3	5.18 × 10^5^	−32.15
M34			31.8	3.28 × 10^3^	32.7	2.69 × 10^3^	−18.10
M35			18.8	3.05 × 10^6^	21.5	1.97 × 10^6^	−35.57
M36			21.0	5.39 × 10^5^	22.6	4.02 × 10^5^	−25.44
N1	NO treatment (Control group)		32.2	4.49 × 10^3^	27.4	1.05 × 10^5^	2238.65
N2			27.7	8.62 × 10^4^	25.7	3.21 × 10^5^	271.88
N3			28.2	6.21 × 10^4^	26.5	1.90 × 10^5^	205.38
N4			24.1	9.17 × 10^5^	21.4	5.40 × 10^6^	488.90
N5			29.8	3.01 × 10^4^	25.9	2.81 × 10^5^	832.57
N6			23.8	1.12 × 10^6^	23.1	1.77 × 10^6^	58.36

* The microwave treatment for each citrus plant was conducted a total of 10 cycles. ** The percentage change of *C*Las titer = (titer of *C*Las after treatment-titer of *C*Las before treatment) ÷ titer of *C*Las before treatment × 100%.

**Table 3 plants-14-02712-t003:** Comparison of differential metabolites in microwave-treated citrus samples and control samples using positive and negative polarity modes of mass spectrometry.

Compared Groups	Num. of Total Ident. *	Num. of Total Sig. **	Num. of Sig. Up ***	Num. of Sig. Down ****
A vs. B	1747	160	74	86
C vs. D		123	31	92
E vs. Fd		174	43	131
G vs. H		95	40	55
J vs. K		59	28	31

* Numbers of total identified metabolites. ** Numbers of total significant differential metabolites. *** Numbers of total significant upregulated metabolites. **** Numbers of total significant downregulated metabolites.

**Table 4 plants-14-02712-t004:** Key differential metabolites shared among comparison groups.

Num.	Polarity Mode (*m*/*z*)	Metabolite Name	Molecular Formula	Compound Type	A vs. B	C vs. D	E vs. Fd	G vs. H	J vs. K	Up or Down
1	+	Beta-caryophyllene	C_15_H_24_	Lipids and lipid-like molecules	0 *	1 **	1	1	1	up
2	−	LPI 16:0	C_25_H_49_O_12_P		1	1	1	1	0	up
3	−	LPI 18:3	C_27_H_47_O_12_P		1	1	0	1	1	up
4	−	2-Methoxyestradiol	C_19_H_26_O_3_		1	1	1	1	1	down
5	−	Bornyl acetate	C_12_H_20_O_2_		1	1	1	0	1	down
6	+	Enoxolone	C_30_H_46_O_4_		1	1	1	1	0	down
7	+	Methylmalonate	C_4_H_6_O_4_	Organic acids and derivatives	1	1	1	1	1	down
8	+	2-Oxoadipic acid	C_6_H_8_O_5_		1	1	1	1	1	down
9	−	5′-S-methyl-5′-thioadenosine	C_11_H_15_N_5_O_3_S	Nucleosides, nucleotides, and analogues	1	1	1	1	1	down
10	−	Gentiopicrin	C_16_H_20_O_9_	Others	1	1	1	1	1	down
11	−	Kaempferol-3-Galactoside-6″- Rhamnoside-3‴-Rhamnoside	C_33_H_40_O_19_		0	1	1	1	1	down
12	−	2-{[(1-benzothiophen-3-ylmethyl)- 3-imino]methyl}phenol	C_16_H_13_NOS		1	1	1	1	0	down
13	+	(6E)-7-(2H-1,3-benzodioxol-5-yl)-1- (piperidin-1-yl)hept-6-en-1-one	C_19_H_25_NO_3_		1	1	1	1	0	down
14	+	4-oxo-4-[(1-phenylethyl)amino]but-2- 5-enoic acid	C_12_H_13_NO_3_		1	1	1	1	0	down
15	+	6-methylpyrimido [4,5-d]pyrimidin-4- 7-amine	C_7_H_7_N_5_		0	1	1	1	1	down

* Not detected. ** Detected.

## Data Availability

The metabolomic data in this study were submitted to the MetaboLights database (https://www.ebi.ac.uk/metabolights/, accessed on 30 May 2025) and are publicly available. The data accession is the following: MTBLS12543.

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
