# Peer review of "Microwave Treatment for Citrus Huanglongbing Control: Pathogen Elimination and Metabolomic Analysis"

_plants, 2025, doi:10.3390/plants14172712_

Round 1

Reviewer 1 Report

Comments and Suggestions for Authors

First, this paper is one of the few truly innovative responses to the HLB epidemic that I have seen. I strongly encourage the authors to pursue this approach. There is one similar paper I know of, from Sri Lanka, in which a group of researchers applied strong electric current to palms affected by phytoplasmas. It apparently cured the palms, at least temporarily. I will try to attach that paper for you, and if that is not allowed, I will send it to the editor. I think you should cite it. Researchers around the world have tried nearly everything that has worked for other pest or pathogen problems, with little success in mitigating HLB. (Some of these, in my opinion, should have been non-starters, but at least now we have real data that they don’t work!) We need something entirely new. I think that experimentation using the resources of the electromagnetic spectrum could be very useful.

Second, evidently, I am unable to obtain an editable version (MS Word, for example) of this manuscript to make edits in the language. I strongly encourage you to find a native speaker of English and provide him or her with a copy that can be edited easily. I would be able and willing to do this for you, but not without a conveniently editable version. Similarly, I might have misinterpreted some things in this paper due to problems understanding your English. If so, please just let the editor know what you intended to say.

Substantive comments:

  1. The final USDA estimate for the current (2024–2025 season) Florida orange crop is 12 million 90 pound boxes. Less than 30 years ago, we produced over 240 million. That is a 95% reduction in yield for the state, mostly due to HLB (citrus greening). Please fix lines 34-37 accordingly.
  2. In section 2.1 (by the way, please fix the title), you tested 400 samples from “the market.” Were these plants for sale in a nursery? If so, first of all, you need much more stringent nursery regulations. There is another problem, though, and that is being sure that plants that test negative in such an environment are actually clean. Our research indicates that we find positive psyllid vectors in a retail venue 9 months before our inspectors find symptomatic plants that will test positive for the pathogens. In the field, the pathogens can hide for several years. The incubation period for this disease can be long and is highly variable. Moreover, the pathogens frequently are undetectable in asymptomatic tissue. Please elaborate how you ensured that your negative plants were truly negative.
  3. In Table 1, the data do not appear to be replicated; however, in line 379 of the methods section, you seem to suggest that you had six replicates. Which is true? Was there one plant for each time and power combination, or were there six?
  4. How did you determine that there were both thermal and non-thermal effects? I suspect you are right, but this should be spelled out better.
  5. In section 2.3, you discuss the variability among treated trees. It could be critical to understand what causes that variability. We know that the incubation period for HLB (the disease) is highly variable. What causes one block to manifest severe disease in a year, and another to persist in a healthy state with undetectable CLas for several years? If we knew why certain plants were able to fight successfully for years, we might be able to solve the whole problem! If we could extend the 6-year incubation period (observed at least twice in Florida) to 15 – 20 years, we could grow a profitable crop.
  6. In table 2, you list (apparently) six treated plants and six untreated plants. Are these all you tested? I think you should test many more in order to have meaningful statistics about the physiology of these plants. Much of this paper deals with metabolomics and biochemistry. I think that first, you should have a foundation demonstrating the efficacy of this novel technique. I think that instead of six plants, you should treat 100 or more. Maybe you would only need 20 or 30 control plants (depending on variability), but you need to treat a lot of them to prove that microwave treatment does eliminate the pathogens, at least temporarily until more vectors arrive. Later papers can begin to sort out why it occurs and explore questions like to one addressed in section 2.3. This is a novel treatment that people will question because it’s different from anything anyone else has tried. It is important to demonstrate beyond any doubt that it works before you do detailed biochemistry. Again, maybe I have misinterpreted Table 2, and if so, please explain that in your response.
  7. I did not find anything in the metabolomics and biochemistry section addressing differences between plants M2 and M5. These two plants showed different responses to the treatment. Did they respond differently in their biochemical profiles?
  8. Table 4 seems to have a template title that does not reflect its content.

In conclusion, I am excited to see novel management ideas for HLB, and I think there is reason to believe that exploring the electromagnetic spectrum could prove useful and even result in a breakthrough. However, if you only tested six plants, I think you should focus now on testing more plants to be sure that the method works. The metabolomics and biochemistry could be subjects of later papers.

Comments on the Quality of English Language

Please see above. I strongly advise soliciting help from a native speaker of English.

Author Response

Comments and Suggestions for Authors

First, this paper is one of the few truly innovative responses to the HLB epidemic that I have seen. I strongly encourage the authors to pursue this approach. There is one similar paper I know of, from Sri Lanka, in which a group of researchers applied strong electric current to palms affected by phytoplasmas. It apparently cured the palms, at least temporarily. I will try to attach that paper for you, and if that is not allowed, I will send it to the editor. I think you should cite it. Researchers around the world have tried nearly everything that has worked for other pest or pathogen problems, with little success in mitigating HLB. (Some of these, in my opinion, should have been non-starters, but at least now we have real data that they don’t work!) We need something entirely new. I think that experimentation using the resources of the electromagnetic spectrum could be very useful.

Response: Thank you very much for providing the reference from Sri Lanka (Siriwardhana, et al., 2014), it is of great significance for demonstrating the use of electromagnetic fields to control plant pathogens. This paper would cite it in lines 301-304 of the discussion section.

Second, evidently, I am unable to obtain an editable version (MS Word, for example) of this manuscript to make edits in the language. I strongly encourage you to find a native speaker of English and provide him or her with a copy that can be edited easily. I would be able and willing to do this for you, but not without a conveniently editable version. Similarly, I might have misinterpreted some things in this paper due to problems understanding your English. If so, please just let the editor know what you intended to say.

Response: The authors had made efforts to revise this paper and re-upload an editable version of MS Word. If there are still any non-standard English expressions, we sincerely ask the editor to help point them out again. Thank you.

Substantive comments: 

Comments 1. The final USDA estimate for the current (2024–2025 season) Florida orange crop is 12 million 90 pound boxes. Less than 30 years ago, we produced over 240 million. That is a 95% reduction in yield for the state, mostly due to HLB (citrus greening). Please fix lines 34-37 accordingly.

Response 1: Thank you for providing the latest data of USDA, we would fix the relevant expressions in lines 36-37.

Comments 2. In section 2.1 (by the way, please fix the title), you tested 400 samples from “the market.” Were these plants for sale in a nursery? If so, first of all, you need much more stringent nursery regulations. There is another problem, though, and that is being sure that plants that test negative in such an environment are actually clean. Our research indicates that we find positive psyllid vectors in a retail venue 9 months before our inspectors find symptomatic plants that will test positive for the pathogens. In the field, the pathogens can hide for several years. The incubation period for this disease can be long and is highly variable. Moreover, the pathogens frequently are undetectable in asymptomatic tissue. Please elaborate how you ensured that your negative plants were truly negative.

Response 2: Firstly, title of section 2.1 had been fixed to “Detection of Huanglongbing citrus”. Secondly, the 400 citrus samples were not purchased directly from “the market” but elected from a nursery with grid and nets structure to prevent cross-infection by psyllid. The citrus samples had been cultivated in the psyllid-proof nursery with separated seedling cups for about 2 years, so there was almost no possibility of false negatives of HLB. The article mentioned “the market” meaning that if those HLB citrus seedlings were sold to the market would causes a great risk of transmission.

Comments 3. In Table 1, the data do not appear to be replicated; however, in line 379 of the methods section, you seem to suggest that you had six replicates. Which is true? Was there one plant for each time and power combination, or were there six?

Response 3: As shown in the methods section, each citrus plant was treated once on a set of six parallel plants. In Table 1, “Citrus Num.” should be modified to “Citrus Group”. For excemple, “Citrus Group 1” meant that six parallel plants were treated with 1500 W, 30 s microwave treatment, and the highest temperature of “Citrus Group 1” was set at 62 ℃ through the temperature control system.

Comments 4. How did you determine that there were both thermal and non-thermal effects? I suspect you are right, but this should be spelled out better.

Response 4: As an electromagnetic wave, the thermal and non-thermal effects of microwave is inherently determined by its wave-particle duality. In the “discussion” section, the following description could be seen: “The thermal effect of microwaves arised from the interaction of the alternating electric field with polar molecules in the citrus tissue, generating heat through molecular friction [31] ”, and “Additionally, microwaves exhibited non-thermal effects, wherein electromagnetic energy alters the charge distribution and transmembrane potential of cell membranes, leading to membrane rupture and lysis [32].” This is a common physical effect of microwave. Whether treated with citrus plants or other organisms, both thermal and non-thermal effects would occur simultaneously.

Comments 5. In section 2.3, you discuss the variability among treated trees. It could be critical to understand what causes that variability. We know that the incubation period for HLB (the disease) is highly variable. What causes one block to manifest severe disease in a year, and another to persist in a healthy state with undetectable CLas for several years? If we knew why certain plants were able to fight successfully for years, we might be able to solve the whole problem! If we could extend the 6-year incubation period (observed at least twice in Florida) to 15 – 20 years, we could grow a profitable crop.

Response 5: We completely agree with the editor's point of view, citrus varieties or plants show different sensitivities to HLB. As described in lines 45-46 of the “introduction” section, researchers are attempting to obtain citrus germplasms resistant to HLB through various methods. In section 2.3, citrus shown different titer change rate of CLas after treatment. We speculate that on the one hand, it was due to the difference in the sensitivity of plants to HLB, and on the other hand, it might be caused by the difference in pathogen concentration before treatment. Here, we chose citrus M2 and M5 as examples. Citrus M2, which had a relatively high initial CLas titer, saw the greatest decrease in titer change rate of CLas after microwave treatment, while conversely, the titer change rate of CLas in citrus M5 was the smallest. This meant that if the initial CLas titer of citrus was very low, it would be extremely difficult to completely eliminate the pathogens through external methods (lines 280-288).

Comments 6. In table 2, you list (apparently) six treated plants and six untreated plants. Are these all you tested? I think you should test many more in order to have meaningful statistics about the physiology of these plants. Much of this paper deals with metabolomics and biochemistry. I think that first, you should have a foundation demonstrating the efficacy of this novel technique. I think that instead of six plants, you should treat 100 or more. Maybe you would only need 20 or 30 control plants (depending on variability), but you need to treat a lot of them to prove that microwave treatment does eliminate the pathogens, at least temporarily until more vectors arrive. Later papers can begin to sort out why it occurs and explore questions like to one addressed in section 2.3. This is a novel treatment that people will question because it’s different from anything anyone else has tried. It is important to demonstrate beyond any doubt that it works before you do detailed biochemistry. Again, maybe I have misinterpreted Table 2, and if so, please explain that in your response.

Response 6: In fact, we tested hundreds of HLB citrus plants to verify the effectiveness of microwave treatment for elimination of CLas. Due to “CLas shows thermal sensitivity and reduced viability at temperatures above 35 ℃ [19]” (lines 49-50), microwave treatment of citrus fruits at 250, 500 W or other power levels could all reduce theCLas titer to a certain extent. In this paper, we only presented the data of 6 representative citrus plants. After microwave treatment, their titer change rate of CLas ranged from 65.03 to 99.83%, which could fully reflect the variation pattern of pathogen concentration in other treated citrus plants. More importantly, after verifying the effectiveness of microwave treatment in eliminating pathogens, we also hoped to study the impact of microwave on the physiological activities of citrus plants. Therefore, we also selected these six treated and six untreated citrus plants for metabolomic detection and analysis. Citrus leaf samples were collected at 0, 24, 48, 72, and 96 hours post-microwave treatment, a total of (6+6)×5=60 samples were tested. We believed that this sample size was sufficient for metabolomic analysis. If we presented more microwave-treated citrus data, the number of samples applied to metabolomic analysis would increase exponentially.

Comments 7. I did not find anything in the metabolomics and biochemistry section addressing differences between plants M2 and M5. These two plants showed different responses to the treatment. Did they respond differently in their biochemical profiles?

Response 7: In the metabolomics and biochemistry section, We mainly compared the differences between microwave-treated citrus and untreated citrus. Therefore, citrus M2 and citrus M5 were classified as Group A (after microwave treatment 0 h), Group C (24 h), Group E (48 h), Group G (72 h) and Group K (96 h) in the data statistics (Table 3, Figure 6 and Figure 7). So this paper did not present anything in metabolomics and biochemistry differences between plants M2 and M5. The reason why these two plants showed different responses to the treatment, please refer to our response to “substantive comments” 5 above, thank you.

Comments 8. Table 4 seems to have a template title that does not reflect its content.

Response 8: We are very sorry that it was our mistake that caused you a misunderstanding. We would fix the title of Table 4 to “Main differential metabolites shared among comparison groups.”

Reviewer 2 Report

Comments and Suggestions for Authors

The paper studied Candidatus on Citrus for their microwave treatment. This is an exciting study. However, some concerns need to be addressed.

1-Why did the authors choose after 90 days to see the result of citrus treating by microwave?

2-Why did the authors not study the phytochemical changes after the microwave testing? Some of the phytochemicals affect the final production, as well as combat other plant pathogens. This must be reflected in the Results and Discussion by the authors. 

Author Response

Comments and Suggestions for Authors

The paper studied Candidatus on Citrus for their microwave treatment. This is an exciting study. However, some concerns need to be addressed.

Comments 1: Why did the authors choose after 90 days to see the result of citrus treating by microwave?

Response 1: Firstly, When the HLB pathogen CLas died after microwave treatment, its DNA still remained in the plant and was gradually degraded. Therefore, it was more accurate to wait for a certain period of time after treatment before detecting the titer of CLas. Secondly, previous researches had proposed that using physical methods to control citrus HLB chose to test the titer of CLas at 90 days after treatment (like references 26, 36). Therefore, the authors believed that this was a widely recognized testing time for CLas. Thirdly, because the incubation period of CLas is very long and the residual pathogen might continue to multiply in the plant, the authors would continue to test the titer of CLas at other times, such as one or two years later, to systematically investigate the long-term change pattern of the pathogen after microwave treatment.

Comments 2: Why did the authors not study the phytochemical changes after the microwave testing? Some of the phytochemicals affect the final production, as well as combat other plant pathogens. This must be reflected in the Results and Discussion by the authors. 

Response 2: Actually, It's a very good suggestion to study the phytochemical changes after the microwave testing. We also believe that certain specificphytochemicals will play an important role in the production of citrus fruits and their exposure to pathogens stress. In this paper, authors employed non-targeted metabolomics methods to investigate the main differential metabolites before and after microwave treatment, identified some phytochemicals that might be related to microwave treatment (Table 4), and discussed their possible effects on the plants' response to microwave stress (lines 348-377). Nevertheless, thank you for the suggestions, we are very motivated to study the targetedphytochemicals of citrus after microwave treatment and explore the effects of these compound on plant growth, fruit yield and HLB infection. However, these studies would require more time and more experiments to verify. Maybe we will report the related results in subsequent articles in 1-2 years.

Reviewer 3 Report

Comments and Suggestions for Authors

This well-written paper with sound and original information will provide a leap in the cultural-physical control of HLB in citrus plants. It is the first step in that direction, but it is a solid step.  I have only suggestions on the format, not on the content.

The most important one is in Figure 4, where it is almost impossible to figure out what the authors want to show. I suggest redrawing the figure and increasing the size so that people can get the appropriate idea.

Please, do the following corrections in the manuscript:  

Almost all citations do not leave a space following the previous word (marked in yellow).

Line 53: Change "causing significant harm to" to "significantly harming."

Lines 61-63: The three scientific names must be in italics.

Line 93: Add a space in "Citruswere."

Line 120: eliminate "the" from "the citrus..."

Line 121: Use "were" instead of "was."

Lines 124, 127, 130, 133, 136. Introducing treatment notation (citrus M2) is confusing; I suggest introducing an early reference to Table 2.

Line 130: Use "control group" instead of "control group citrus."

Line 133: Add an "s" to "citru N1."

Line 269: Delete a "t" in "tstem."

Line 232: Remove an extra space in "citrus and CLas".

Lines 300-3002: Reword the following phrase: "This penetrating of microwave heating mechanism ensures simultaneous and uniform temperature elevation across both external and internal tissues, including the phloem, thereby significantly enhancing elimination efficiency of CLas". Suggested new phrase: "This penetrating microwave heating method ensures even temperature increase in both external and internal tissues, including the phloem, significantly improving the efficiency of CLas elimination".

Line 320: Substitute "This study focused our analysis..." using "This study focused the analysis..."

Line 369. Add a colon at the end.

Lines 378-380. Rewrite the phrase "To tset the tolerance of citrus to microwave: microwave power of 250 W, 500 W, 1000 W, 1500 W, microwave time of 15 s, 20 s, 25 s, 30 s; each plant was treated once and set up 6 parallel plants". Suggested new phrase: "To set the tolerance of citrus to microwave: microwave power levels of 250, 500, 1000, and 1500 W; microwave times of 15, 20, 25, and 30 s; each plant was treated once on a set of six parallel plants".

Lines 381-383: Rewrite the phrase "To tset the elimination of CLas by microwave: microwave power of 500 W, microwave time of 20 s, stopped microwave until the citrus displayed room temperature, repeat the treatment for 20 s, repeat the operation 10 times". Suggested new phrase: "o set the elimination of CLas by microwave: microwave power of 500 W, microwave time of 20 s, stopped microwave until the citrus displayed room temperature, repeat the treatment for 20 s, and repeat the operation 10 times".

Lines 384-385: Rewrite the phrase "Placed the citrus treated with microwave and the control group citrus in the seedling shed, cultivated under the same conditions". Suggested new Phrase: "The citrus trees were microwave-treated; the control group was left in the seedling shed and cultivated under the same conditions".

Line 388: Use "citrus plant." instead of only "citrus."

Line 401: Add a space in "95 ℃for..."

Lines 408-411: Rewrite the phrase "A standard curve was generated by plotting the logarithm of plasmid copy number against corresponding CT values: y=3.5063x+7.4913, y represents CT, x corresponds to log(DNA concentration), R2=0.9992, enabling absolute quantification of CLas in experimental samples". Suggested new phrase: "A standard curve was drawn based on the positive control template DNA concentration and CT value. The 1171-bp CLas 16S rDNA fragment was cloned into the pUC57 plasmid, generating a 3887-bp recombinant plasmid (designated pUC-CLas16S) as a positive control template."

Line 414: Eliminate the word "Precisely."

Line 416: Substitute "followed by addition of..." with "adding." 

Lines 443-444: Substitute "group separation, ariable..." with "group separation, variable..."

Lines 511-512: Add a space in "citrusderived..."

Lines 400, 573-574: Reference number 45 is not cited in the paper, and citation 46 does not have a reference. Please revise if citation 46 is 45.

Comments on the Quality of English Language

A series of phrases must be rewritten for clarity. I add suggested phrases to do so.

Author Response

Comments and Suggestions for Authors

This well-written paper with sound and original information will provide a leap in the cultural-physical control of HLB in citrus plants. It is the first step in that direction, but it is a solid step.  I have only suggestions on the format, not on the content.

The most important one is in Figure 4, where it is almost impossible to figure out what the authors want to show. I suggest redrawing the figure and increasing the size so that people can get the appropriate idea.

Response: Thank you very much for your suggestion, Figure 4 had been increased to a larger size. 

Please, do the following corrections in the manuscript:  

Comments 1: Almost all citations do not leave a space following the previous word (marked in yellow).

Response1 : We had done all these corrections according to your suggestion.

Comments 2: Line 53: Change "causing significant harm to" to "significantly harming."

Response 2: We had done the correction according to your suggestion.

Comments 3: Lines 61-63: The three scientific names must be in italics.

Response 3: We had done these three correction according to your suggestion.

Comments 4: Line 93: Add a space in "Citruswere."

Response 4: We had done the correction according to your suggestion.

Comments 5: Line 120: eliminate "the" from "the citrus..."

Response 5: We had done the correction according to your suggestion.

Comments 6: Line 121: Use "were" instead of "was."

Response 6: We had done the correction according to your suggestion.

Comments 7: Lines 124, 127, 130, 133, 136. Introducing treatment notation (citrus M2) is confusing; I suggest introducing an early reference to Table 2.

Response 7: We had added an early introduction to Table 2 in lines 127-129.

Comments 8: Line 130: Use "control group" instead of "control group citrus."

Response 8: We had done the correction according to your suggestion.

Comments 9: Line 133: Add an "s" to "citru N1."

Response 9: We had done the correction according to your suggestion.

Comments 10: Line 269: Delete a "t" in "tstem."

Response 10: We had done the correction according to your suggestion.

Comments 11: Line 232: Remove an extra space in "citrus and CLas".

Response 11: We had done the correction according to your suggestion.

Comments 12: Lines 300-3002: Reword the following phrase: "This penetrating of microwave heating mechanism ensures simultaneous and uniform temperature elevation across both external and internal tissues, including the phloem, thereby significantly enhancing elimination efficiency of CLas". Suggested new phrase: "This penetrating microwave heating method ensures even temperature increase in both external and internal tissues, including the phloem, significantly improving the efficiency of CLas elimination".

Response 12: We had reworded the phrase according to your suggestion.

Comments 13: Line 320: Substitute "This study focused our analysis..." using "This study focused the analysis..."

Response 13: We had done the correction according to your suggestion.

Comments 14: Line 369. Add a colon at the end.

Response 14: We had done the correction according to your suggestion.

Comments 15: Lines 378-380. Rewrite the phrase "To tset the tolerance of citrus to microwave: microwave power of 250 W, 500 W, 1000 W, 1500 W, microwave time of 15 s, 20 s, 25 s, 30 s; each plant was treated once and set up 6 parallel plants". Suggested new phrase: "To set the tolerance of citrus to microwave: microwave power levels of 250, 500, 1000, and 1500 W; microwave times of 15, 20, 25, and 30 s; each plant was treated once on a set of six parallel plants".

Response 15: We had rewrited the phrase according to your suggestion.

Comments 16: Lines 381-383: Rewrite the phrase "To tset the elimination of CLas by microwave: microwave power of 500 W, microwave time of 20 s, stopped microwave until the citrus displayed room temperature, repeat the treatment for 20 s, repeat the operation 10 times". Suggested new phrase: "o set the elimination of CLas by microwave: microwave power of 500 W, microwave time of 20 s, stopped microwave until the citrus displayed room temperature, repeat the treatment for 20 s, and repeat the operation 10 times".

Response 16: We had rewrited the phrase according to your suggestion.

Comments 17: Lines 384-385: Rewrite the phrase "Placed the citrus treated with microwave and the control group citrus in the seedling shed, cultivated under the same conditions". Suggested new Phrase: "The citrus trees were microwave-treated; the control group was left in the seedling shed and cultivated under the same conditions".

Response 17: We had rewrited the phrase according to your suggestion.

Comments 18: Line 388: Use "citrus plant." instead of only "citrus."

Response 18: We had done the correction according to your suggestion.

Comments 19: Line 401: Add a space in "95 ℃for..."

Response 19: We had done the correction according to your suggestion.

Comments 20: Lines 408-411: Rewrite the phrase "A standard curve was generated by plotting the logarithm of plasmid copy number against corresponding CT values: y=3.5063x+7.4913, y represents CT, x corresponds to log(DNA concentration), R2=0.9992, enabling absolute quantification of CLas in experimental samples". Suggested new phrase: "A standard curve was drawn based on the positive control template DNA concentration and CT value. The 1171-bp CLas 16S rDNA fragment was cloned into the pUC57 plasmid, generating a 3887-bp recombinant plasmid (designated pUC-CLas16S) as a positive control template."

Response 20: We had rewrited the phrase according to your suggestion.

Comments 21: Line 414: Eliminate the word "Precisely."

Response 21: We had done the correction according to your suggestion.

Comments 22: Line 416: Substitute "followed by addition of..." with "adding." 

Response 22: We had done the correction according to your suggestion.

Comments 23: Lines 443-444: Substitute "group separation, ariable..." with "group separation, variable..."

Response 23: We had done the correction according to your suggestion.

Comments 24: Lines 511-512: Add a space in "citrusderived..."

Response 24: We had done the correction according to your suggestion.

Comments 25: Lines 400, 573-574: Reference number 45 is not cited in the paper, and citation 46 does not have a reference. Please revise if citation 46 is 45.

Response 25: We had done the correction of reference number according to your suggestion.

Round 2

Reviewer 1 Report

Comments and Suggestions for Authors

Dr. Lu:

The English in this manuscript still needs a LOT of improvement. It is very possible that the apparent flaws in design of these experiments are due to poor English, not actual experimental lapses. I think it should be two papers. The really novel discovery of microwave treatment for HLB seems lost in the details of metabolomics. Based on the comments you sent me, they really do have enough treated trees to make claims of efficacy. Instead of picking 6 representative plants (necessary for the detailed molecular studies), they need to report the results of all the tests to demonstrate efficacy of the microwave treatment.

Comments on the Quality of English Language

The English still needs a lot of improvement. I suspect that the apparent experimental flaws are due to language issues, but before the language is cleaned up, I cannot tell. It would be a shame to lose the report of this innovative research due to poor English. Please find a native speaker with subject familiarity who can help with this.

Author Response

Comments and Suggestions for Authors

The last time I reviewed this paper, I commented that this is one of the first truly innovative responses to the global HLB epidemic. I still agree wholeheartedly with that, and the information you supplied in the responses to my comments makes me even more optimistic that publication of these findings will be very possible.

That said, I think this should be two papers, not one. The first one should report the discovery that microwave treatment suppresses Candidatus Liberibacter asiaticus, the pathogen associated with HLB. It should include a discussion of all the plants you treated.

I worried that you had only tested your microwave methods on six plants, far too few to claim the technique works, but in your response to my comment number 6, you said “in fact we tested hundreds of HLB citrus plants to verify the effectiveness of microwave treatment of elimination of CLas.” Did you report this somewhere else? (I think not, or you would have cited it.) It seems that you are mentioning this major discovery as an afterthought to your molecular testing, whereas it really is the main event! If this really works, there will be entrepreneurs who will figure out how to scale it up and make it work in the field. When someone discovered that heat treatment would set back the HLB, there were American entrepreneurs who made steamer tents to treat citrus canopies. Unfortunately, the steam tenting made all the leaves fall off, and psyllids attacked the new growth immediately (!). Besides, the steamers did not affect CLas in the roots. The remedy (along with the tractor-mounted steamers) was short lived. Your treatment, on the other hand, sets back the disease, treats the roots as well as the canopy, and does not damage the tree. If you can write a paper that shows everyone that this works, there will be many people devising ways to make it work at scale. This paper should give details about the times and power used to suppress CLas without killing or damaging the plants. Include your photos, the treatment parameters, a detailed description, along with photos, of the machine you used to generate the microwave radiation, etc. Do give details about cultivars tested to show that the treatment is independent of cultivar. Please do cite the similar work in Sri Lanka. Ever since I read about their experiments, I wondered if something similar might work for CLas. According to your data, it does. This paper will be cited a lot and should stand alone, uncomplicated by the metabolomics information.

The second paper about the metabolomics information should be published after but citing the first one. A long time ago, I did several years of extensive research on barley yellow dwarf virus and its vectors. A wise older colleague suggested to me that the information had too many topics. He told me how to break it into two papers, published together, and citing each other. It worked well. I think something similar should be done with this research. I realize that many institutions, probably including yours, highly value molecular “high tech” research. However, the truly innovative discovery in your case is that microwave treatment will suppress HLB. This doesn’t detract from the excellence of your molecular laboratory, but in this case I strongly advise that you publish the major discovery, and then explain it with a stand-alone molecular paper that also will make your laboratory shine.

Response: Dear reviewer,

Thank you very much for your suggestion. You recommended dividing this work into two separate papers: one that focuses on the suppression of Candidatus Liberibacter asiaticus (CLas) through microwave treatment, and the other that addresses the metabolomics information.

We sincerely appreciate the reviewer's thoughtful suggestion. After careful consideration, we have decided to maintain this work as a single integrated manuscript. Our main reason for this decision is that it allows us to simultaneously demonstrate the efficacy of microwave treatment in reducing CLas titers while also highlighting its regulatory effects, including potential negative impacts, on citrus plant growth. This comprehensive presentation will better help readers evaluate the advantages and limitations of microwave treatment for Huanglongbing (HLB) control.

To address the reviewer's concerns, we have made the following revisions: (1) Expanded data on CLas suppression: We have included more detailed results and discussions regarding the changes in CLas titers under various microwave treatment conditions. This strengthens the aspect of our study. (2) Streamlined metabolomics analysis: We have condensed the metabolomics section to emphasize the key findings. This ensures a balanced presentation of both microbiological and physiological outcomes.

We believe the revised manuscript clearly communicates the dual effects of microwave treatment on pathogen elimination and plant metabolism, providing a more comprehensive understanding for researchers in this field.

Thank you again for your review.

Here is our response to your comments and suggestions:

Comments 1. We cannot say that HLB is “caused” by CLas, because technically Koch’s postulates have not been completed. I tend to use “associated with” instead.

Response 1: Thank you for pointing this out. We agree with this comment. Therefore, we have accordingly changed “caused by” to “associated with” in line 9 and line 31.

Comments 2. The material in your comments that explains the thermal and non-thermal effects of microwave treatment is good and should go in the introduction to the first paper. Maybe I have missed something, but I still don’t see how you ascertained that both these effects were present. Would it take microscopic evaluation of CLas cell tissues to see the disruption of membranes?

Response 2: We agree with your comments. We have moved the following text from the discussion section to the introduction section, specifically in lines 63-71:

"The thermal effect of microwaves is caused by the interaction between the alternating electric field and polar molecules in citrus tissue, which generates heat through molecular friction. Because of their penetrating nature, microwave heating ensures a uniform temperature distribution both internally and externally. Additionally, microwaves produce non-thermal effects, where electromagnetic energy alters the charge distribution and transmembrane potential of cell membranes, potentially leading to membrane rupture and lysis. The high-frequency oscillation of microwaves can also disrupt macromolecules such as proteins and nucleic acids, resulting in cell death."

Comments 3. Thanks for fixing the tragic statistics about Florida. Maybe your innovative discovery will be part of the answer.

Response 3: Thank you for providing the latest data of USDA, again.

Comments 4. Under “Results,” the English still needs to be fixed. Maybe the title of 2.1 should read “Detection of Candidatus Liberibacter asiaticus in Citrus.” We always need to distinguish between the disease (a sick plant with symptoms) and the pathogen in question. Chinese also makes this linguistic distinction.

Response 4: Thank you for pointing this out. We agree with this comment. As a result,, we have updated the title of section 2.1 to “Detection of Candidatus Liberibacter asiaticus in citrus” (line 88).

Comments 5. I am still confused about the source of your experimental plants. Were these 400 plants for sale in a “market,” and you detected the disease in 64 of them after you bought them? I am also unclear about how you kept your negative plants negative. Were they grown from seed? The HLB pathogens can hide insidiously for a long time.

Response 5: Thank you for highlighting this issue. As mentioned in comments 4, “We always need to distinguish between the disease (a sick plant with symptoms) and the pathogen”. We completely agree with this statement. Consequently, we have revised our results concerning "HLB disease" and "the CLas pathogen," and we have retracted our earlier comments regarding the HLB-negative citrus plants.

Comments 6. Line 134: I don’t understand “titer change rate decreased by 99.83%. What is titer change rate?

Response 6: We are very sorry for causing you any misunderstanding. We have corrected the phrase “titer change rate of CLas” to “The rate of change in CLas titer”. The rate of change in CLas titer is calculated as follows: The rate of change in CLas titer= (Titer of CLas after treatment - titer of CLas before treatment) ÷ titer of CLas before treatment × 100%. For example, when we say “the rate of change in CLas titer decreased by 99.83%,” it is calculated as: [(3.44E+04)-(2.01E+07)] ÷ (2.01E+07) × 100%= -99.83%. Thank you for your understanding.

Comments 7. In the metabolomics section, I don’t understand Table 3 at all. Why are results repeated twice? I also don’t understand the explanations at the bottom. Please have a native English speaker with subject familiarity fix this.

Response 7: Thank you for pointing this out. Indeed, the results in Table 3 were not repeated twice. The “ion mode of m/z +” refers to the number of differential metabolites detected in the positive polarity mode of the mass spectrometer, while the “ion mode of m/z -” refers to those detected in the negative polarity mode. To enhance clarity, we have combined the data from both the positive and negative polarity modes in Table 3. Additionally, we have revised the explanations at the bottom of Table 3 for better understanding.

Comments 8. I don’t understand lines 275-282.

Response 8: We sincerely apologize for any misunderstanding caused. As a result, we have removed these comments.

Comments 9. Thanks for citing the work in Sri Lanka. I am not sure that the observed effect was due to treating a phytoplasma rather than true bacteria (if that’s what you meant). I’m also not sure that voltages alone explain differences in dosages required. The wavelength of the current involved certainly also matters. This section also needs English editing.

Response 9: Thank you for providing the reference regarding the use of electric current to control Weligama Coconut Leaf Wilt Disease. We have updated the citation in lines229-230 of this paper (References [33]).

Comments 10. In general, the English still needs a lot of editing. Please find a native speaker who has familiarity with the subject matter to help with this. You have made a major discovery. In order to make the impact it deserves, it should be published in excellent English.

Response 10: The English version has been extensively re-edited, and all changes are highlighted. We sincerely hope to receive your valuable comments and suggestions again.

Round 3

Reviewer 1 Report

Comments and Suggestions for Authors

July 2025

This paper has improved a lot since version 2. I like the fact that you have expanded the section on the microwave treatments. Here are a few more comments.

I am going to keep the "major revision recommendation," because I want to see many more of the plants tested for microwave mitigation of HLB included in the report. Without that issue, it would require only editing, or minor revisions. (See points 3-5 below.)

  1. In your introduction, the second paragraph deals with many of the strategies that people are using to deal with HLB. However, it neglects to mention that none of these management tools are working very well. Others are still experimental and certainly not ready for prime time. This paragraph is a good place to emphasize that so far, nothing we have tried is economically viable. (If it were, Florida would not have lost 95% of its crop.) In the next paragraph, you state, “Due to the limited effectiveness of current HLB management strategies…” Here is where you say that you are trying microwave treatments. This is not an emerging technology, at least not yet. You are the first!
  2. If known, you might want to discuss in the introduction about the various components of the microwave treatment. For one thing, “microwave” is a huge part of the electromagnetic spectrum. I see now that you used 2,450 MHz, in the longer wavelength WIFI band. The wavelength might be important. Additionally, watts is a function of volts, current, and impedance. I wonder what varying any of these parameters would do. Are there any studies about the effects on biological materials of various wavelengths in the GHz bands? What about varying volts and current? Why did you settle on this wavelength and 1,500 watts? What was the voltage?
  3. Probably most important, I still don’t see the results for all the diseased plants that you tested. In your previous response, you said that you tested hundreds of plants. Thirty-six is better than the six presented in version 1, but why not use results from all your plants? In any case, you need to state clearly how many plants are involved in each experiment. If you really did test hundreds, you should have plenty of plants. In section 2.3, please state the numbers of plants tested, and include as many as possible.
  4. In Table 1, how many plants were in each group?
  5. Were different cultivars tested? Is this phenomenon independent of cultivar? This should be discussed if a variety of cultivars were used.
  6. Here’s something I wonder. (This point is for follow up, not this paper, although it could be mentioned in the discussion.) We know from experiments with the potato/CLso system that the bacteria may be concentrated in the lower stem of the plant. We also know that CLas can hide for years prior to symptom expression. Maybe it too is hiding in the lower trunk. We know that roots can be positive when you can’t find any bacteria in the canopy. Microwave antennas can be highly directional. I wonder if you aimed the dose at the roots and the part of the tree just above the soil if you would get good results in the field. This also would minimize exposure to microwave radiation for the applicator. Similarly, the people in Sri Lanka shocked only the lower parts of the palms, with efficacious results (and the best sampling strategy for palm phytoplasmas is to drill a core sample in the trunk). This is just something to think about and try later.
  7. Table 2: Rate of change implies a time variable – per second, per week, etc. You are measuring percent change, not a rate of change over a specified time interval (see also line 229). This is a language problem, not a substantive one, I think.
  8. Lines 315-320 (new material) need English editing.
  9. In your final conclusions, I think that the future research should focus first on optimizing microwave treatment for efficacious field use. You should test various wavelengths, combinations of voltage and current, directional antennas, parts of the plant that should be treated, safety issues for applicators, etc. After you find something that works optimally, then do more metabolomics, not the other way around. Assessing the physiological responses will take a lifetime, and we don’t have that long to figure out how to grow citrus in an endemic HLB environment. Optimize your treatment parameters to give the farmers something to try first, always being mindful of safety issues surrounding microwave radiation.

Specific corrections:

First, there is still quite a bit of general editing to do, especially with the new material. Please have a native speaker with an editable version fix these things. Below are a few things I noticed.

Line 59ff: Consider “Antimicrobial activity results from both thermal effects and non-thermal mechanisms, with the latter primarily due to disruption of cellular structures. The thermal effect of microwave treatment is due to the interaction between the alternating electric field and polar molecules within living tissue…”

Line 64: “heating ensures uniform temperature…”

Line 65: “exhibits,” not exhibited. You are talking here about immutable physical phenomena, not a particular experiment, so use present tense.

Line 88: “screened,” not “detected”

Line 93: “citrus plants, some of which exhibited noticeable…”

Line 127 at the end: “were found to be…”

Table 1 title: citrus trees, not citrus fruits (You did not test orange fruits.)

Line 229: “Table 2 presents the percent change in CLas titer following treatment.”

Line 239: “for the pathogen…”

You are almost there! I think that you need some more language clean-up, and an even more thorough presentation of the plants you tested with microwave treatment.

I hope to get a reprint of this paper when it comes out so that I can advocate for more experiments here in Florida with electric current treatments for dealing with fastidious prokaryotes. I think this could have a lot of promise.

Comments on the Quality of English Language

The English still needs  improvement, especially the new material. 

Author Response

This paper has improved a lot since version 2. I like the fact that you have expanded the section on the microwave treatments. Here are a few more comments.

I am going to keep the "major revision recommendation," because I want to see many more of the plants tested for microwave mitigation of HLB included in the report. Without that issue, it would require only editing, or minor revisions. (See points 3-5 below.)

  1. In your introduction, the second paragraph deals with many of the strategies that people are using to deal with HLB. However, it neglects to mention that none of these management tools are working very well. Others are still experimental and certainly not ready for prime time. This paragraph is a good place to emphasize that so far, nothing we have tried is economically viable. (If it were, Florida would not have lost 95% of its crop.) In the next paragraph, you state, “Due to the limited effectiveness of current HLB management strategies…” Here is where you say that you are trying microwave treatments. This is not an emerging technology, at least not yet. You are the first!

Response 1: Thank you very much for your suggestion. We have made necessary revisions in the second and third paragraphs of the introduction section (line 59 and line 61).

  1. microwave treatment. For one thing, “microwave” is a huge part of the electromagnetic spectrum. I see now that you used 2,450 MHz, in the longer wavelength WIFI band. The wavelength might be important. Additionally, watts is a function of volts, current, and impedance. I wonder what varying any of these parameters would do. Are there any studies about the effects on biological materials of various wavelengths in the GHz bands? What about varying volts and current? Why did you settle on this wavelength and 1,500 watts? What was the voltage?

Response 2: Firstly, 2,450 MHz is an universally accepted worldwide frequency band designated by the International Telecommunication Union (ITU) that is available for use by industrial, scientific, and medical equipment. While other common frequency bands, such as 915 MHz, may vary between countries. Secondly, water molecules in biological cells resonate intensely in the 2,450 MHz microwave alternating electric field, which converts microwave energy into heat energy. Additionally, 2,450 MHz microwaves can penetrate deeper into the interior of plants. Finally, the magnetron technology associated with 2,450 MHz microwaves is well-established and cost-effective, making it suitable for large-scale applications in agriculture and industry. For these reasons, the authors chose to use 2,450 MHz microwaves for our research. Unfortunately, we currently only have access to 2,450 MHz microwave system, and the laboratory’s working voltage is limited to 220 V, preventing us from exploring other microwave frequencies and voltages. Our microwave system offers various power options, ranging from a minimum of 150 W to a maximum of 5100 W. We tested the tolerance of citrus fruits under four microwave power settings: 250 W, 500 W, 1250 W, and 1500 W. Our findings indicated that exposure to 1500 W of microwaves resulted in the death of citrus plants. As a result, we opted to use 250 W and 500 W as the microwave energy levels for pathogen elimination. Importantly, the authors acknowledge that different wavelengths within the microwave frequency band may have varying effects on biological materials. However, it is essential to first obtain microwave devices that operate at specific frequencies, such as 915 MHz. This will necessitate a new experimental cycle and additional verification of samples.

  1. Probably most important, I still don’t see the results for all the diseased plants that you tested. In your previous response, you said that you tested hundreds of plants. Thirty-six is better than the six presented in version 1, but why not use results from all your plants? In any case, you need to state clearly how many plants are involved in each experiment. If you really did test hundreds, you should have plenty of plants. In section 2.3, please state the numbers of plants tested, and include as many as possible.

Response 3: As shown in Section 2.3 and Table 2, the percent change of CLas titer in 36 citrus plants varied between 18.10% and 99.83% after undergoing 10 cycles of microwave circulation treatment. In contrast, other citrus samples exposed to different microwave conditions—such as higher power levels, longer treatment durations, or increased cycles (e.g., 20 cycles, 30 cycles)—suffered severe damage or even death. This was due to the microwave radiation energy exceeding the tolerance levels of the citrus plants. While we conducted microwave treatment experiments on over 100 citrus plants, the percent change of CLas titer among those that were severely damaged or dead were not statistically analyzed. This is because these extreme microwave treatment conditions are not practical for large-scale applications and could lead to further destruction of citrus orchards. Therefore, this paper focuses solely on the 36 citrus plants, which we believe constitutes a sufficient sample size to demonstrate the effects of microwave treatment.

On the other hand, in order to achieve a 100% microwave elimination efficiency for the CLas and to ensure that no hidden bacteria reemerge after years of treatment, as you mentioned in comments 2 and 9, we indeed need to conduct citrus processing experiments using various microwave frequencies, voltages, and currents. Additionally, the microwave antenna's radiation area should be able to simultaneously process both the above-ground and underground parts of the plants. This will require a new microwave device equipped with a highly intelligent control system to ensure the controllability and adjustability of these microwave parameters. We believe that with the introduction of more effective microwave technology, we will be able to conduct further processing experiments and provide more data on percent change of CLas in citrus samples in the future.

  1. Table 1, how many plants were in each group?

Response 4: There were 6 plants in each group. This means that Table 1 shows a total of 96 plants in 16 groups were subjected to the microwave tolerance test (lines 321-326) .

  1. Were different cultivars tested? Is this phenomenon independent of cultivar? This should be discussed if a variety of cultivars were used.

Response 5: In this study, we selected only one variety of citrus for the microwave treatment experiment.

  1. Here’s something I wonder. (This point is for follow up, not this paper, although it could be mentioned in the discussion.) We know from experiments with the potato/CLso system that the bacteria may be concentrated in the lower stem of the plant. We also know that CLas can hide for years prior to symptom expression. Maybe it too is hiding in the lower trunk. We know that roots can be positive when you can’t find any bacteria in the canopy. Microwave antennas can be highly directional. I wonder if you aimed the dose at the roots and the part of the tree just above the soil if you would get good results in the field. This also would minimize exposure to microwave radiation for the applicator. Similarly, the people in Sri Lanka shocked only the lower parts of the palms, with efficacious results (and the best sampling strategy for palm phytoplasmas is to drill a core sample in the trunk). This is just something to think about and try later.

Response 6: This is an excellent suggestion. In the future research, we will explore the use of microwaves to treat the roots of specific soil sections. Our primary objective is to determine whether the penetrating power of microwaves can completely eliminate all CLas found in the stems, leaves, and roots of citrus plants.

  1. Table 2: Rate of change implies a time variable – per second, per week, etc. You are measuring percent change, not a rate of change over a specified time interval (see also line 229). This is a language problem, not a substantive one, I think.

Response 7: Thank you for pointing this out. We agree with this comment. The "Rate of change in CLas titer" in Table 2 has been revised to "percent change of CLas titer".

  1. Lines 315-320 (new material) need English editing.

Response 8: Section 4.2 (Materials and Methods) has been revised and edited for clarity.

  1. In your final conclusions, I think that the future research should focus first on optimizing microwave treatment for efficacious field use. You should test various wavelengths, combinations of voltage and current, directional antennas, parts of the plant that should be treated, safety issues for applicators, etc. After you find something that works optimally, then do more metabolomics, not the other way around. Assessing the physiological responses will take a lifetime, and we don’t have that long to figure out how to grow citrus in an endemic HLB environment. Optimize your treatment parameters to give the farmers something to try first, always being mindful of safety issues surrounding microwave radiation.

Response 9: We fully agree with your opinion. It is necessary to conduct experiments with different microwave processing conditions and different treatment parts of the plants. Moreover, while utilizing the microwave equipment in the laboratory, we would monitor for any microwave radiation leakage to ensure the safety of laboratory personnel. If microwaves prove to be an effective treatment for HLB, it will also be crucial to maintain microwave leakage within safe limits when employed outdoors. Accordingly, we have made the necessary modifications to the conclusions section of the paper based on your suggestions.

Specific corrections:

First, there is still quite a bit of general editing to do, especially with the new material. Please have a native speaker with an editable version fix these things. Below are a few things I noticed.

Line 59ff: Consider “Antimicrobial activity results from both thermal effects and non-thermal mechanisms, with the latter primarily due to disruption of cellular structures. The thermal effect of microwave treatment is due to the interaction between the alternating electric field and polar molecules within living tissue…”

Response : Thank you very much for your suggestion. We have revised the sentence in lines 63-67.

Line 64: “heating ensures uniform temperature…”

Response : Thank you for pointing this out. We have revised the sentence in line 68.

Line 65: “exhibits,” not exhibited. You are talking here about immutable physical phenomena, not a particular experiment, so use present tense.

Response : Thank you for highlighting this issue. We have revised this word in line 69.

Line 88: “screened,” not “detected”

Response : Thank you for highlighting this issue. We have revised this word in line 92.

Line 93: “citrus plants, some of which exhibited noticeable…”

Response : Thank you for pointing this out. We have revised the sentence in lines 97-98.

Line 127 at the end: “were found to be…”

Response : Thank you for your suggestion. We have revised the sentence in line 133.

Table 1 title: citrus trees, not citrus fruits (You did not test orange fruits.)

Response : Thank you for pointing this out. We have revised the title of Table 1.

Line 229: “Table 2 presents the percent change in CLas titer following treatment.”

Response : Thank you for highlighting this issue. We have revised the sentence in lines 236-237.

Line 239: “for the pathogen…”

Response : Thank you for your suggestion. We have revised the sentence in line 247.

Round 4

Reviewer 1 Report

Comments and Suggestions for Authors

Microwave treatment to treat HLB version 4 review

This paper is much better this time around, and you have answered my questions satisfactorily. I have found a few editorial corrections again and will list those. Please, before this goes to press, have a native speaker give it a good read with an editable version (Microsoft Word or equivalent) in hand.

In your response to my questions, you wrote a paragraph (edited a bit) that I think should be incorporated in some form into the paper:

“To achieve a 100% microwave elimination efficiency for CLas and to ensure that no hidden bacteria reemerge after treatment, further work is needed, including experiments using various microwave frequencies, voltages, and currents. Additionally, the microwave antenna's radiation pattern is a variable that can be investigated. Ideally, the equipment should be able to treat both the above-ground and underground parts of the plants simultaneously. This will require development of new microwave devices equipped with highly intelligent control systems to ensure the adjustability of microwave parameters and coverage. We believe that with the introduction of more effective microwave technology, we will be able to conduct further experiments and provide more data on efficacy. This includes reduction in CLas titer and, ultimately, prolonged health and productivity of citrus trees, including good yields and high-quality fruit.

[Ed. note: This can’t be a one-time treatment anyway, because there still will be psyllids. Treatment frequency is another possible subject of investigation. This management strategy will need to be accompanied by pretty good control of D. citri.]

In your discussion and conclusions, I still think there should be more emphasis on practical considerations. We do this stuff because we love citrus, and we want to see it thrive again. We want to see farmers make a good living and take pride in growing an abundance of tasty fruit. It seems to me that sometimes we allow our tools to take precedence over our social goals. All the amazing tools we have at our disposal today (electronic equipment, AI, molecular biology, etc.) provide opportunities to serve the public, and in our case, crops and those who produce them. We in agricultural science must never lose sight of that.

I disagree with the last sentence (final conclusion) in this article for reasons stated above. Metabolomic analysis is a powerful tool, but it doesn’t determine whether microwave treatment can be an economically viable part of HLB management. That question is determined foremost by whether it works, and whether treated trees thrive to produce delicious fruit. That is the first goal. In your discussion, plans for future research might include treating a lot more plants, experimenting with other cultivars to see if this treatment is independent of cultivar, field experiments with different kinds of directional antennas and various other frequencies and dosage parameters, safety experiments, parts of the trees to treat for maximum suppression of CLas, etc. Your goal is to manage HLB so farmers make money, citrus trees don’t suffer, and people have delicious citrus fruit to enjoy.

Editorial stuff and a few minor comments:

Line 15: Consider “99.83% reduction in CLas titer. Non-targeted metabolomic analysis identified…” The % sign assumes percent change.

Line 18: effectively suppressed

Lines 58–59: Consider “Due to the limited effectiveness of current HLB management strategies, we tested microwave treatment to see if it could be a promising physical approach.”

Lines 232–233: Consider “This variation might be attributable to the differences…”

Line 301: Please list the name of the cultivar you used.

Lines 320–327 still have some syntax issues. Consider “Microwave treatments to reduce or eliminate CLas in citrus were given in six groups….” Continuing with line 323 “In each of the microwave experiments we completed one cycle of microwave treatment, then paused the machine to allow the plants to return to room temperature. Once the trees had cooled, we restarted the machine and repeated the process for another cycle. Total treatment required ten cycles, with cooling pauses between each successive treatment.” [Ed. note: How long did it take to complete the required 10 cycles?]

Line 388–389: …and we also will examine the effects of microwave treatment on CLas in roots in the field. [Ed. note: I think you should stick with HLB for this paper. After you optimize HLB mitigation, you probably will have money, lab infrastructure, and expertise to branch out!]

Comments on the Quality of English Language

Please have a native speaker give this one last read, with an editable version (Microsoft Word or equivalent) in hand to make changes. It is greatly improved, but would still benefit from one last look.

Author Response

Comments and Suggestions for Authors

Microwave treatment to treat HLB version 4 review

This paper is much better this time around, and you have answered my questions satisfactorily. I have found a few editorial corrections again and will list those. Please, before this goes to press, have a native speaker give it a good read with an editable version (Microsoft Word or equivalent) in hand.

In your response to my questions, you wrote a paragraph (edited a bit) that I think should be incorporated in some form into the paper:

“To achieve a 100% microwave elimination efficiency for CLas and to ensure that no hidden bacteria reemerge after treatment, further work is needed, including experiments using various microwave frequencies, voltages, and currents. Additionally, the microwave antenna's radiation pattern is a variable that can be investigated. Ideally, the equipment should be able to treat both the above-ground and underground parts of the plants simultaneously. This will require development of new microwave devices equipped with highly intelligent control systems to ensure the adjustability of microwave parameters and coverage. We believe that with the introduction of more effective microwave technology, we will be able to conduct further experiments and provide more data on efficacy. This includes reduction in CLas titer and, ultimately, prolonged health and productivity of citrus trees, including good yields and high-quality fruit.

Response 1: Thank you very much for your suggestion. We have incorporated this paragraph in the “Discussion” section of the paper (lines 268-278).

[Ed. note: This can’t be a one-time treatment anyway, because there still will be psyllids. Treatment frequency is another possible subject of investigation. This management strategy will need to be accompanied by pretty good control of D. citri.]

Response 2: We fully agree with your opinion. We will conduct long-term tracking experiments and at the same time pay attention to control D. citri (lines 319-320).

In your discussion and conclusions, I still think there should be more emphasis on practical considerations. We do this stuff because we love citrus, and we want to see it thrive again. We want to see farmers make a good living and take pride in growing an abundance of tasty fruit. It seems to me that sometimes we allow our tools to take precedence over our social goals. All the amazing tools we have at our disposal today (electronic equipment, AI, molecular biology, etc.) provide opportunities to serve the public, and in our case, crops and those who produce them. We in agricultural science must never lose sight of that.

Response 3: We support your idea. Consequently, we have added a paragraph in the “Discussion” section to highlight the author's commitment to safeguarding the citrus industry and improving the well-being of farmers (lines 315-321).

I disagree with the last sentence (final conclusion) in this article for reasons stated above. Metabolomic analysis is a powerful tool, but it doesn’t determine whether microwave treatment can be an economically viable part of HLB management. That question is determined foremost by whether it works, and whether treated trees thrive to produce delicious fruit. That is the first goal. In your discussion, plans for future research might include treating a lot more plants, experimenting with other cultivars to see if this treatment is independent of cultivar, field experiments with different kinds of directional antennas and various other frequencies and dosage parameters, safety experiments, parts of the trees to treat for maximum suppression of CLas, etc. Your goal is to manage HLB so farmers make money, citrus trees don’t suffer, and people have delicious citrus fruit to enjoy.

Response 4: Thank you very much for pointing this out. We share your vision of benefiting the citrus industry and supporting farmers. The authors believe it is crucial for future studies to focus on various microwave processing conditions, different citrus cultivars, and different parts of the plant. Additionally, investigating fruit yield and quality is essential. Therefore, in the “Discussion” (lines 268-278) and “Conclusions” (lines 415-416, and lines 418-421) sections, we have highlighted the key points and goals for future work.

Editorial stuff and a few minor comments:

Line 15: Consider “99.83% reduction in CLas titer. Non-targeted metabolomic analysis identified…” The % sign assumes percent change.

Response 5: We appreciate you bringing this issue to our attention. Corrections have been made (line 15).

Line 18: effectively suppressed

Response 6: Thank you for pointing out this error. We have revised it to "effectively suppressed" (line 18).

Lines 58–59: Consider “Due to the limited effectiveness of current HLB management strategies, we tested microwave treatment to see if it could be a promising physical approach.”

Response 7: We agree with your opinion and have made the necessary revisions as suggested (lines 60-61).

Lines 232–233: Consider “This variation might be attributable to the differences…”

Response 8: We appreciate this suggestion and have made the necessary revisions accordingly (line 237) .

Line 301: Please list the name of the cultivar you used.

Response 9: We have already listed the cultivar name: "navel orange (Citrus sinensis Osb. var. brasiliensis Tanaka)" (lines 324-325).

Lines 320–327 still have some syntax issues. Consider “Microwave treatments to reduce or eliminate CLas in citrus were given in six groups….” Continuing with line 323 “In each of the microwave experiments we completed one cycle of microwave treatment, then paused the machine to allow the plants to return to room temperature. Once the trees had cooled, we restarted the machine and repeated the process for another cycle. Total treatment required ten cycles, with cooling pauses between each successive treatment.” [Ed. note: How long did it take to complete the required 10 cycles?]

Response 10: Thank you for pointing out these issues. We have made the necessary revisions according to your instructions (lines 345-346, and lines 347-351). [The citrus plants must cool from the 32-52 ℃ temperature reached after microwave heating to room temperature (23 ± 2 ℃), which usually takes 5 to 15 minutes. Therefore, a complete 10 cycles microwave treatment would require approximately 50 to 150 minutes.]

Line 388–389: …and we also will examine the effects of microwave treatment on CLas in roots in the field. [Ed. note: I think you should stick with HLB for this paper. After you optimize HLB mitigation, you probably will have money, lab infrastructure, and expertise to branch out!]

Response 11: Thank you for your suggestion. We have made corresponding revisions to the “Conclusions” section (lines 415-416, and lines 418-421). [We will continue to apply for research funding in the future to conduct further experiments on citrus microwave treatment and to develop a microwave processing device with additional functions and parameters.]

Comments on the Quality of English Language

Please have a native speaker give this one last read, with an editable version (Microsoft Word or equivalent) in hand to make changes. It is greatly improved, but would still benefit from one last look.

Response 12: We have invited a native English speaker to read the article. We believe that most readers will be able to clearly understand its content. Once again, we would like to express our gratitude for your meticulous work in reviewing this article.
